# Study of pulsed electrostatic field (PESF) in the perfusion of peripheral tissues: Microangiopathy, nutrition and quality of perceiver life

**Rossella Liani** [1]*, **Sara La Torre**[1], **Valentina Liani**[2], **Angela Melchiorre**[2], **Danilo D'Ettorre**[2], **Romina Tripaldi**[1], **Stefano Lattanzio**[1], **Rossano Di Luzio**[2], **Mauro Coli**[3], **Carlo Velussi**[4]

**1** Department of Medicine and Aging, CAST_Center for Advanced Studies and Technology, University G. d'Annunzio of Chieti-Pescara, Chieti, Italy, **2** Nephrology and Dialysis Unit, "S. Massimo" Hospital, Penne, Italy, **3** Department of Economics, Institute of Statistics, University of Chieti, Chieti, Italy, **4** Retired Associate Professor of Physiology, University of Padova, Padova, Italy

* rossellaliani@yahoo.it

## Abstract

Microangiopathy compromises the structural and functional integrity of organs and tissues in patients with type II diabetes mellitus (T2DM) negatively affecting the perceived quality of life. Nitric oxide (NO) is a multifunctional signalling molecule, acting as a vasodilator, neurotransmitter, and modulator of inflammatory processes. Patients with type II diabetes mellitus and chronic kidney disease, controlled from glycaemic status, were treated or not with pulsed electrostatic field (PESF) cycles to evaluate effect on the perfusion of peripheral tissues. Everyone was monitored for the metabolic profile, and we tested circulating NO with a commercial enzyme immunoassay kit. In addition, we tested the perceived quality of life of patients before/after a PESF cycle using a questionnaire. Patients treated with PESF were improved circulating NO levels, significant changes in systolic and diastolic blood pressure, heart rate and were more homogeneous for their metabolic profile. The questionnaire showed also a marked improvement in the perceived quality of life. The use of pulsed electrostatic fields has allowed us to observe an improvement in the metabolic, psychological, and clinical profile in patients with T2DM and chronic kidney disease whose pathological profile is strongly compromised.

## Introduction

The prognosis for patients with type 2 diabetes mellitus (T2DM) has changed significantly over the course of a few decades thanks to the continuous technological and pharmacological innovations as a result of a fruitful collaboration between industry and scientific-healthcare world. Despite the progress in terms of survival, the perceived quality of life in old patients with T2DM, however, is compromised by several chronic complications. Intensive glucose control, after a follow-up of 15 years, doesn't produce substantial differences among the groups on cardiovascular outcome dependent on diabetes, it is not associated with significant benefits

**Data Availability Statement:** All relevant data are within the paper.

**Funding:** I have to disclose research grant support from Akern. The funders had no role in the study design; collection, analysis, and interpretation of data; writing of the paper; and/or decision to submit for publication. The funders had no served or currently serve on the editorial board of the journal to which they are submitting, had no acted as an expert witness in relevant legal proceedings, and had no sat or currently sit on a committee for an organization that may benefit from publication of the paper. The authors received no specific funding for this work.

**Competing interests:** I have to disclose research grant support from Akern. The funders had no role in the study design; collection, analysis, and interpretation of data; writing of the paper; and/or decision to submit for publication. The funders had no served or currently serve on the editorial board of the journal to which they are submitting, had no acted as an expert witness in relevant legal proceedings, and had no sat or currently sit on a committee for an organization that may benefit from publication of the paper. This does not alter our adherence to PLOS ONE policies on sharing data and materials.

for the quality of life [1] and it does not prevent T2DM typical chronic complications [2, 3]. In elderly patients there are chronic pain and sensory or motor paraesthesia [4], vegetative neuropathy, peripheral neuropathy, sense organs failure [5], sarcopenia [6] and loss of cognitive functions in relation to cerebral neuropathy [7]. Neurovegetative and somatic complications can coexist [8], interfere with the subjective and affective experience and undermine the relationship between patients and their environment [9, 10].

In T2DM the microvascular disorder is systemic and affects the integrity and the functions of the biological heritage [11]. In elders with T2DM and chronic kidney disease (CKD), without macrovascular precedents, microangiopathy effects the early atrophy of the brain matter of the frontal and temporal lobes and it affects memory, the executive functions and ultimately the behaviour [12]. The addition of vasoactive medications to those antidepressant does not improve mood in patients with depressive syndrome [13–15]. Recent studies have reported interesting clinical effects after administration of negative electric charge (PESF) or after exposure to an electromagnetic field (PEMF) in patients with chronic disease, that cannot be completely attributable to a simple vasodilatation. PESF potentiates microcirculation autonomous motility or "vasomotion" [16], it dissolves or inhibits rouleaux formation [17, 18], increases the baseline of circulating nitric oxide (NO), delivery of oxygen ($O_2$) to the peripheral tissue, basal metabolic rate [19] and, finally, accelerates ulcers healing in patients with T2DM [20]. PEMF dissolves erythrocyte aggregates [21], increases the diameter of microcirculation vessels in T2DM, accelerates blood velocity in cutaneous microcirculation and improves healing of skin wounds [22]. In rat model PEMF stimulates the expression of mRNA $HCN_1$/$HCN_2$ in peripheral nerves with experimental chronic neuropathy [23], limits the outcomes of experimentally induced strokes, acts on neuroinflammation and on circulating NO level [24] and, when administering extremely low-frequency pulses, activates antioxidant defence mechanisms in human osteoblasts [25].

## Materials and methods

### Patient population

Our study, which was made possible by the generous work of doctors and nurses, was carried out in five years (2009–2014). It involved male and female patients aged over 60 years controlled from glycaemic point of view, in periodic haemodialysis for at least six months. The means to be employed (PESF and questionnaire), the use conditions and the purpose have been fully described to all the candidates. Family members have been involved in the case of dependent patients. In no case the drug intake at home and the rhythm of dialysis were modified. Thirty individuals agreed to participate in the study and were assigned to group A or B with "simple random system", alternately according to their order of arrival. Group A (mean age 75.70±12.36 years) was composed of 13 women and 12 men (HbA1c = 6.76±0.80%) and group B (mean age 74.93±8.52 years) was composed of 12 women and 13 men (HbA1c = 6.64 ± 0.86%). Group A (treated) was subjected to a real PESF (NewHealtH9000, Akern / RJL Systems) treatment, while group B (control) to a simulated PESF treatment. During the simulated sessions we did not administer negative charges in group B. Each "cycle of PESF", real and simulated, was performed before dialysis, each session lasted 30 minutes and was repeated three times a week, every other day, for a total of 13 times.

We used impedance tests (BIA-101, Akern / RJL Systems) to establish the intensity of PESF treatment based on each patient's cell mass index [19] and to monitor the nutritional status of all patients at the end of the PESF cycle.

Blood samples were taken before the start of the dialysis session, at the end of the dialysis and 30 minutes after the end of the purifying session on the first purifying treatment of the

week (after the long interdialytic period) at baseline and repeated after the PESF cycle (13 sessions) before starting the dialysis session, at the end and after 30 minutes of dialysis.

The sampling for the electrophoresis control of serum proteins was carried out only before the dialysis session, pre and post PESF.

The results were obtained through observational and retrospective clinical reports using the electronic archive present in our Operating Unit in accordance of STROBE criteria [26]. We excluded patients with major arteries occlusion, valued with doppler test and patients who used: polyphenols, antioxidants for their effects on the cellular signal; phytotherapics (eg. Centella asiatica extracts) for their known effects on endothelial cells; anabolic drugs for their effects on metabolism; and psychotropic drugs.

All participants provided written informed consent and had the opportunity to withdraw from the study without providing any explanation. The protocol was approved by the General Health Director (DS1687_24.02.2010).

## Nitric oxide assay

Circulating NO blood levels were tested using ELISA commercial kit (#KGE001, R&D). The method is a spectrophotometric that uses Greiss reaction which is based on enzymatic conversion of nitrates to nitrites due to nitrate reductase [27, 28].

## Questionnaire

As the vasodilatation in some psychotic manifestations is unsatisfactory [29], we decided to evaluate, on the basis of questionnaires, the effects on the perceived quality of life in a group of elderly patients before and after exposure to PESF, assuming that the effects on microcirculation, including effects on motility or "vasomotion", rouleaux breakdown and/or inhibited formation, increased circulating NO level and increased delivery of $O_2$, induce systemic effects which cannot be achieved only by vasodilatation. Individuals with evident microcirculation deficiency, patients with skin wounds, have been chosen for the study. Skin wounds are very common in patients with T2DM and CKD, they depend on the efficiency of microcirculation, and, in the late stages of CKD, they occur more frequently and may affect the quality of life [30].

The questionnaire forms were downloaded from the website [31] and then adapted to our operational reality without changing the purpose and respecting Gordon's suggestions, who considers the patient a "unique subject" despite its complexity [32]. The monitoring was supposed to ensure changes in self-assessed health in relation to objective limitations imposed by a wide variety of conditions, including perception and management of their own health, food freedom in respect of food patterns, the level of physical activity, cognition and perception capacity, the sleep-wake rhythm and social relationships, which are certainly compromised in case of subjective limitations [33–35]. In addition to the interview the operating model included observation, physical examination, access to the medical records and, in case of doubt, an interview with a family member. The pain that patients had before and after the PESF cycle was assessed by simple scale in which the maximum intensity was 10 and the minimum intensity was 0.

We profiled that an improved tissue perfusion could have effects on artificial blood clearance and on metabolism. To verify the first effects, we checked the azotaemia before, at the end and 30 minutes after the end of the dialysis session (on the first day of dialysis of the week). To determine the second effects, we checked body weight (Kg), phase angle (detected with impedance analysis), azotaemia, blood creatinine, uric acid levels, serum calcium,

phosphatemia, hemoglobinemia, haematocrit, globular volume, percentage of albumin (taken from serum protein electrophoresis) and NO, running tests between the dialysis days.

## Statistical analysis

Univariable comparisons between groups were performed by χ2 tests or Mann-Whitney U or Spearman rank correlation test. Only 2-tailed p<0.05 was considered statistically significant. The data analysis was generated using SPSS Statistic software, Version 25.0 for IOS. The homogeneity of the groups was checked using Andrew's curve. This system extracts from the original variables, according linear and hierarchical combinations, the contribution to the total variance [36]. Andrews curves which have been constructed with the first 5 principal components (CP) equal to 92% of the total variance. Principal components: they are extracted, by means of orthogonal rotation, from the original variables according to linearized combinations and according to their contribution to the total variance. Each curve represents a single individual (group A, pre and post PESF; group B, pre and post simulated PESF). Common laboratory tests (azotaemia, creatinine, uricemia, haemoglobin and albumin) were used for each curve.

## Results and discussion

The effects on metabolism in group A and B are represented in Fig 1, curve of Andrews [36] where each curve represented a single patient and were generated using common laboratory parameters tests (azotaemia, creatinine, uricemia, haemoglobin and albumin), which are usually used in the clinic in patients with CKD. As can been seen, in group B the dispersal of the curves in quadrant POST is not different from that in quadrant PRE, while in the case of group A, in quadrant POST, a representative bundle of curves shows an evident homogeneity compared to those represented in quadrant PRE. As the original variables, in both groups, relate to the uremic status, it can be considered that PESF could have caused, only in group A, appreciable variations of the metabolic status in nearly all subjects examined. Only in group A the individuals communicate to feel less severe pain and therefore to benefit from the treatment received.

In group treated (A) the azotaemia levels, before *vs.* after the PESF cycle respectively, were pre-dialysis 145.81±24.9 mg/dl *vs.* 142.56±21.1 mg/dl, p = ns; post-dialysis 48.94±9.4 mg/dl *vs.* 44.88±4.1 mg/dl, p = 0.09 and after 30 minutes from the end, 58.00±9.7 mg/dl *vs.* 51.00±4.1 mg/dl, p = 0.013.

In the control group (B) the azotaemia levels, before *vs.* after the simulated PESF cycle respectively, were pre-dialysis 147.00±21.4 *vs.* 147.35±21.5 mg/dl, p = ns; post dialysis azotaemia 54.05±12.4 mg/dl *vs.* 50.00±9.9 mg/dl, p = ns and after 30 minutes, 60.05±11.7 mg/dl *vs.* 57.45±11.5 mg/dl; p = ns.

The azotaemia level checked 30 minutes after the end of the depurative session is significantly reduced compared to the value measured before PESF cycle, only in group A (Group A pre *vs.* post, p = 0.01). Moreover, in group A the same value is statistically lower than in the control group B (post: group A *vs* B, p = 0.02).

The results obtained through the questionnaire in group A and B are shown in Table 1. Indicators in order to document the perceived quality of life before and after PESF suggest that the administration of pulsed negative charges has positive effects only in group A, where the opinion on the physical functions, the limits due to the disease, the limitations due to emotional problems and, in general, on the disease has considerably improved.

In addition, though not explicitly investigated, all have reported that they had improvements in performing common daily activities (household duties and spare time) and that waking hour, resting hours and the perception of the quality of sleep had changed for the better.

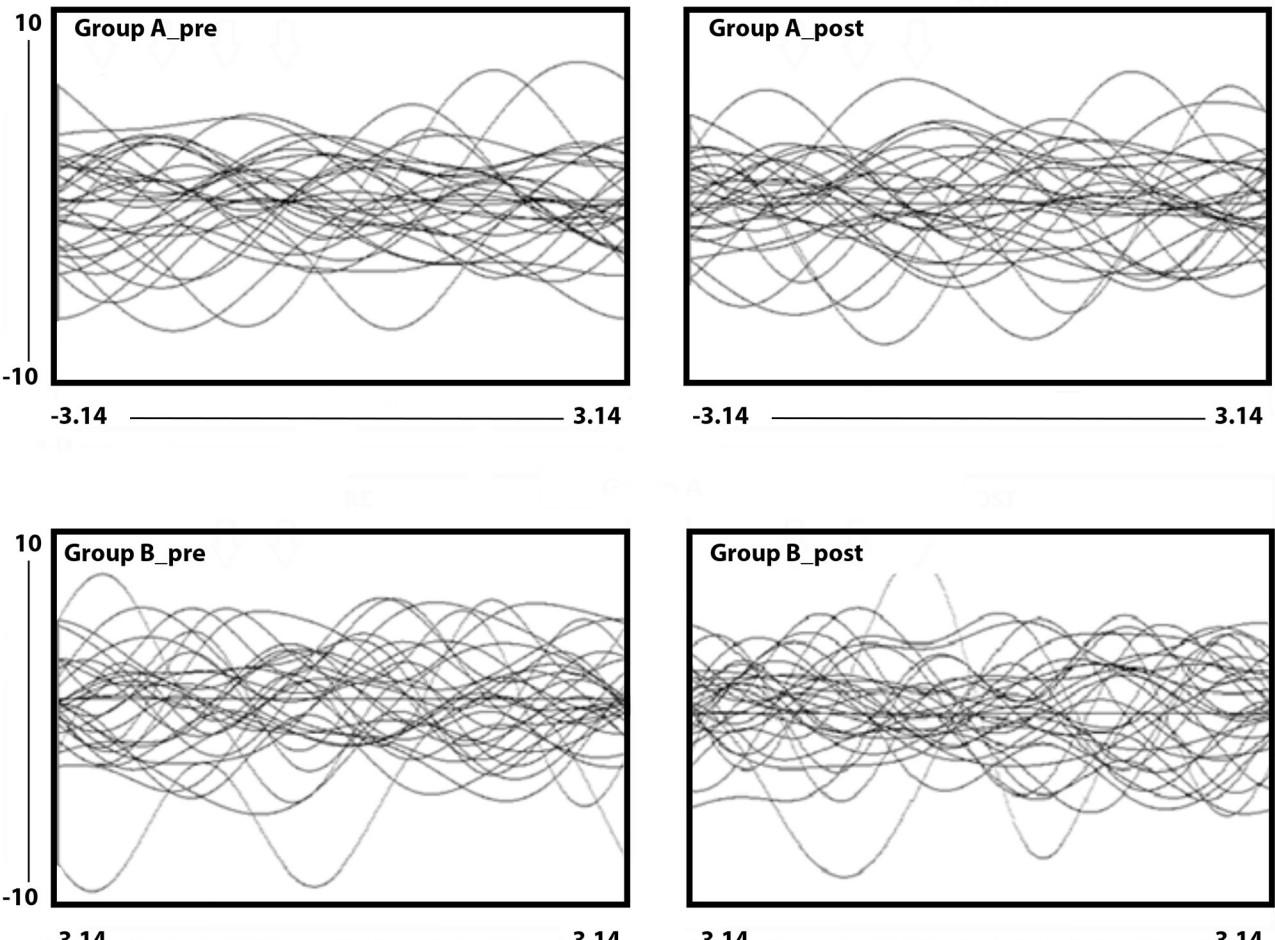

**Fig 1. Graphical representation of the PESF effects on some laboratory tests through the Andrews curves which have been constructed with the first 5 principal components (CP) equal to 92% of the total variance.** Each curve represents a single individual (group A, pre and post PESF; group B, pre and post simulated PESF). Common laboratory tests (azotaemia, creatinine, uricemia, haemoglobin and albumin) were used for each curve. The end of the graphs is: along the X axis: -3.14 and 3.14 and along the Y axis: -10 and +10.

**Table 1. Results obtained by the health questionnaire SF-31 standard in group A and B.** The results show a significant improvement in the group as regards the physical function, the limits imposed by the disease, emotional problems and overall, the opinion about the disease. Significance was calculated by χ2 tests.

| Table 1 | Group A | | | Group B | | |
|---|---|---|---|---|---|---|
| | PRE | POST | p | PRE | POST | p |
| Physical functioning | 58 | 74 | 0.0320 | 53 | 45 | ns |
| Role limitation due to physical health | 23 | 79 | 0.0001 | 37 | 38 | ns |
| Role limitation due to emotional problems | 38 | 80 | ns | 49 | 44 | ns |
| Energy/Fatigue | 53 | 66 | ns | 68 | 60 | ns |
| Emotional well being | 59 | 74 | ns | 70 | 70 | ns |
| Social functioning | 57 | 78 | ns | 68 | 64 | ns |
| Pain | 59 | 69 | ns | 74 | 73 | ns |
| General Health | 47 | 65 | 0.0136 | 58 | 55 | ns |

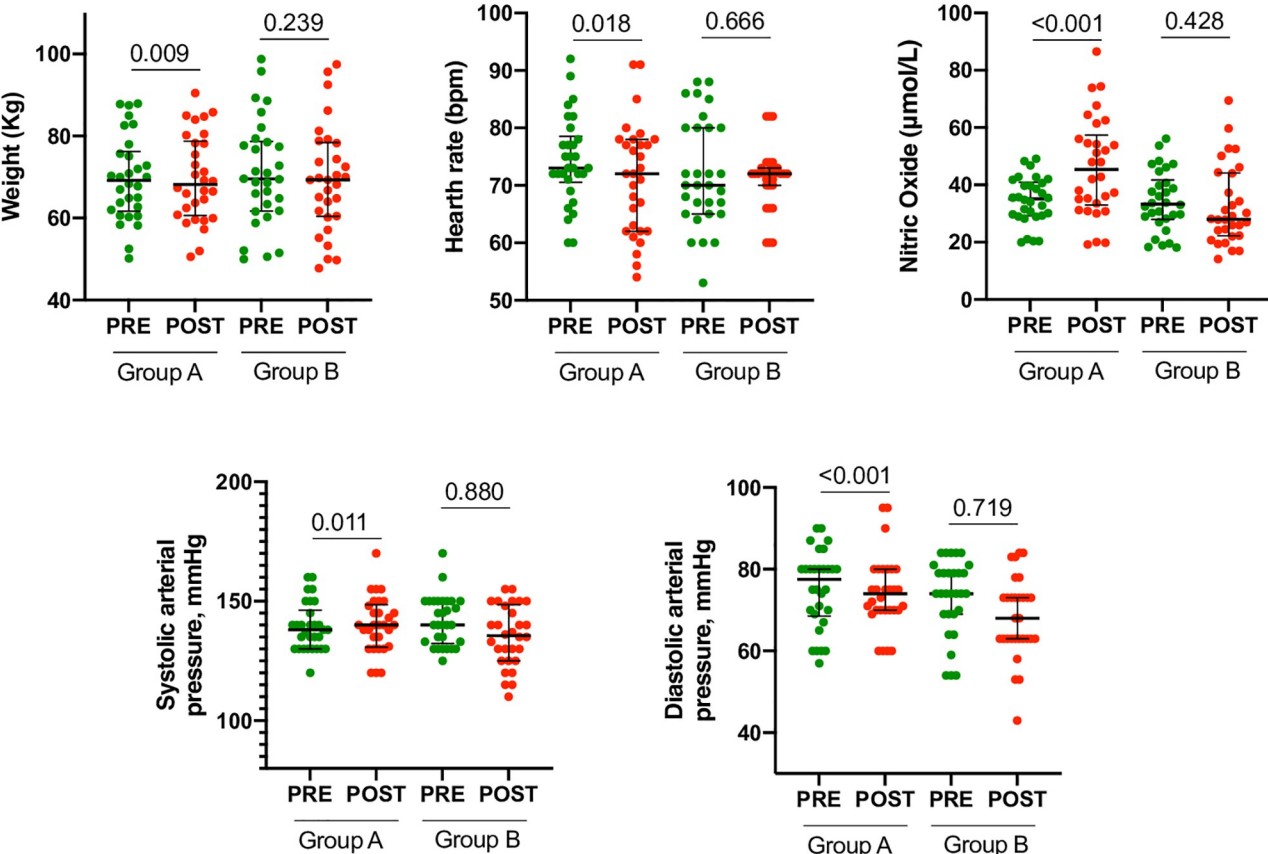

**Fig 2. Clinical and laboratory characteristics in Group A e B, pre and post, PESF or not.** Body weight, systolic and diastolic pressure, heart rate changes significantly reduced, while Nitric Oxide increased only in Group A post treatment. Significance was calculated by Mann-Whitney U test.

Changes in body weight, systolic pressure, diastolic pressure, and heart rate are set out in Fig 2. As can be seen, only in group A, statistically significant variations are recorded. We have illustrated illustrates changes in circulating NO levels in group A, before and after a real PESF treatment, and in group B, before and after a simulated treatment. As can be observed there has been a statistically significant increase of circulating NO level only in group A.

The impedance test did not show significant changes in the biological heritage in the two groups at the end of treatment for fat, fat-free mass, cell mass, muscle mass and hydration (total water and extracellular). On the contrary, the phase angle is slightly but statistically increased in group A (pre = 4.99±1.2, post 5.48±1.1; p = 0.019) and not changed in group B (pre = 4.97±1.3; post 5.06±1.8; p = ns).

The phase angle expresses the integrity of cell membranes and is therefore indirectly related to cellular activity in Fig 3. The increase in the percentage of albumin in group A alone suggests that PESF may have improved the synthesis or release of albumin, but it is not excluded that PESF has improved vascular permeability and reduced albumin passage in the tissues.

## Conclusions

In the design phase of the study, we were aware of the extreme difficulty to be able to document neurological clinical benefits in elderly patients with T2DM and CKD after a single PESF cycle for two good reasons: the unavoidable clinical differences between an individual and the

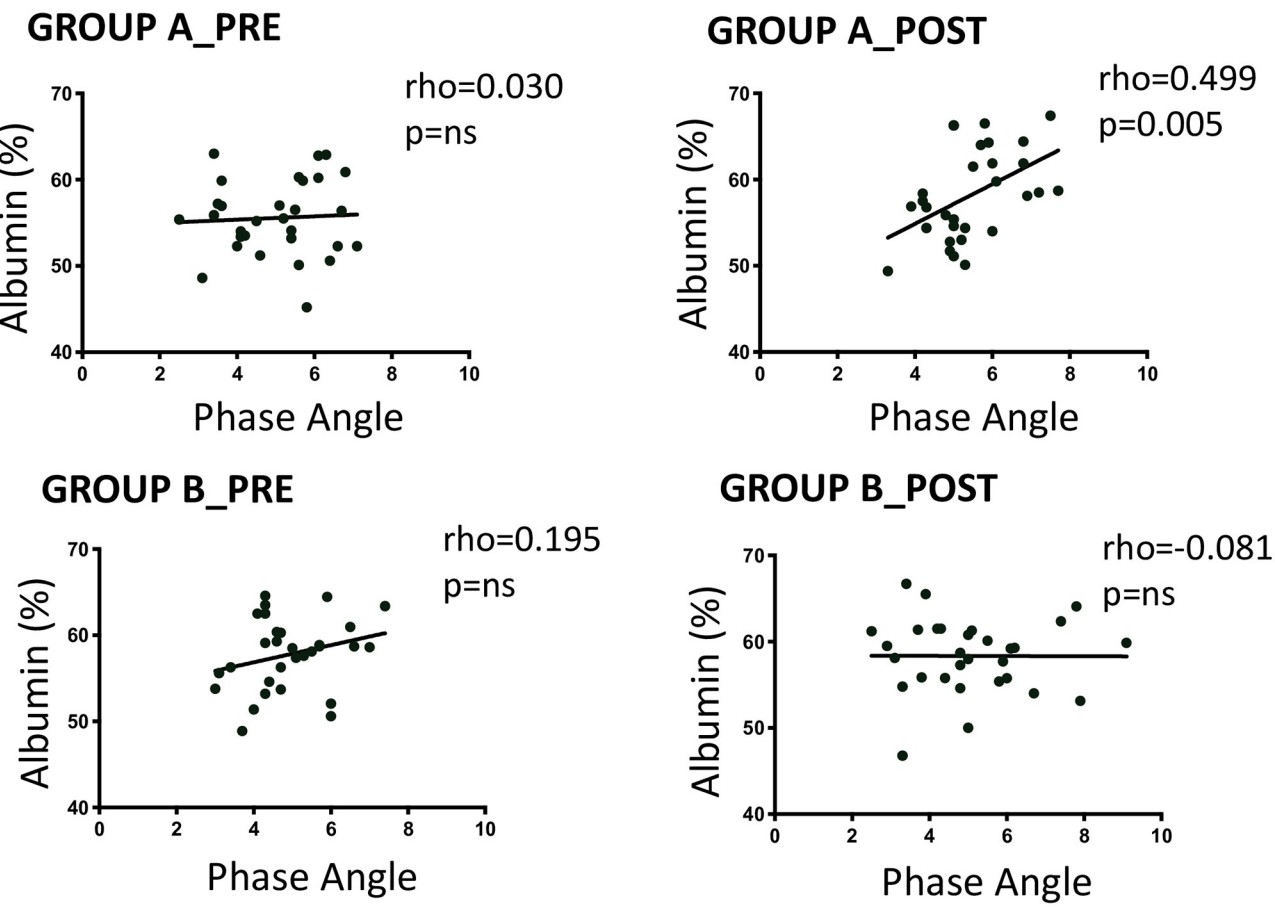

**Fig 3. Correlation between baseline levels of phase angle (PA) and albumin (%) in Group A e B, pre and post, PESF or not.** Spearman correlation coefficient and p-value are reported.

other of the same group, and the absence of direct effects of a electrostatic field of low intensity on central neurological tissue [37].

Microcirculation is a set of vessels between the arterial and the venous system and its efficiency is entrusted to the structural and functional integrity of endothelial cell [38]. The evaluation of the microcirculation functions, endothelium-employee, is a common prospective assessment method in the development and progression of chronic cardio-metabolic and/or kidney disease [39].

As we get older microcirculation undergoes significant structural and functional modifications that are the origin of many clinical implications [40]. The process of involution is accelerated in conjunction with chronic disease: vascular network appears rarefied, the vessels are tortuous, the endothelial glycocalyx is deconstructed and the endothelial synthesis of some protein and NO is reduced [39, 41].

The correlation between neuronal cognition and microcirculation dysfunction has been documented in the elderly population without any alteration of macrocirculation's [42]. Perceived quality of life improvements has been documented by the processing of data collected by a questionnaire before and after the PESF cycle. The unsatisfactory effect of vasoactive medicines, added to antidepressants [15], suggests that vasoconstriction is not presumably the only or the main physical-pathological mechanism. Some people have been reporting for a long time the effects of erythrocyte aggregates or rouleaux on $O_2$ transportation [43].

At present, there are no known molecular targets that relate to the accelerating involution of microcirculation and no medications, that can prevent it, are available. In the elderly population with T2DM and CKD the anti-aging recommendations are like those of other patients and the coexistence of physical and pharmacological fragility [44] are an insurmountable obstacle that cannot be modified even with multidisciplinary approaches [45]. The role of microcirculation is not limited to a passive supply of nutrients [46], but also to an active elimination of uremic toxins [47]. We believe that the extent of accumulation can locally affect the functional integrity and the biological heritage.

Dietary restrictions lead to limited benefits [48] and, when excessively restricted, may favour hypovitaminosis and endothelial dysfunction [49].

Physical limitations negatively affect the intellectual dysfunction and alter the ability of self-management in everyday life and the perceived quality of life [50]. In young chronic renal patients with T2DM vigorous and prolonged physical exercises lead to appreciable effects on physical performance, they lead to self-esteem, with positive effects on anxiety, fatigue and sleep disorder, and ultimately on social reintegration [51]. In elderly patients the possibilities of physical conditioning are limited [52] and early neurocognitive deterioration may encourage self-exclusion and accentuate a sedentary lifestyle and loneliness. It is therefore important to stimulate the physical function in order to minimize the risk of loss of independence and to improve the quality of life in chronic patients. Many have documented these improvements which are attributed to better performance where microcirculation plays an important role [53, 54]. Some have described improvements in athletes undergoing stressful muscle exercises and attributed the results to effects of PESF on microcirculation [55].

Assisting patients with T2DM and CKD is the result of a portentous technological and pharmacological pathway. Clinical results, on the rehabilitation level, are sometimes incredible, and we can't understand the differences of result even when they are significantly different between a patient and another. Different metabolic conditions could be at the origin of the differences, affect the results and, in chronic patients, affect the perception of the quality of life. We have shown that PESF improves some of the rates of perceived health and makes more homogeneous a group of patients that was previously inhomogeneous as regards the metabolic profile.

The phase angle (PA) is universally considered a sensitive indicator of well-being that varies with changes in health. The reliability of PA in sport requires further investigation but is considered a sensitive indicator of health and integrity of muscle tissue [56, 57], malnutrition [58] and frailty [59]. Our study shows that in elderly hemodialysis patients PESF, after just over a month, did not change the biological heritage (fat mass, lean mass, cell mass and muscle mass) but affected the functional heritage as suggested by the increase in angle of phase.

Patients with CKD have a high incidence of micro and macroangiopathy. The reduced availability of endogenous NO, consequent to endothelial dysfunction, can lead to peripheral vasoconstriction, impaired haemostasis, enhanced platelet activation and above all impaired permeability of the vascular wall [60]. It has been shown that, where there is a certain degree of vascular reactivity, this is often accompanied by changes in electrical charges both on the endothelium and on the surface of the erythrocytes [61]. In patients with chronic kidney disease, hypercholesterolemia (LDL), hyperglycemia, inflammation, over-selling and altered shear stress, can damage and degrade the glycocalyx with loss of negative electrical charges and induce endothelial dysfunction, aggravating the risk of atherosclerosis [62]. From the literature data we therefore hypothesized that the circulating NO level is a good marker of endothelial damage. In addition, vascular compromise is the leading cause of morbidity and mortality in CKD patients [63]. Dialysis techniques do not help depressed endothelial activity [64]. In our opinion, endothelial dysfunction is reversible, and the result affects clinical

parameters dependent on vascular resistance [16] and PESF, inducing an increase in NO could affect at least some of the compromised endothelial functions including new vascular resistance with beneficial effects on healing. of difficult wounds in diabetic patients [20]. The use of electric fields of the order of those generated by the bloodstream influences the essential biology of endothelial cells, and the release of NO [65] suggests that the negative electrical charges administered benefited the biosynthetic activity of endothelial NO (eNOS) and presumably erythrocyte (rNOS) NO [65]. Weight reduction and increased levels of NO only in patients who have undergone treatment with PESF confirm its role in the reduction of abdominal obesity [66] in the influences endothelial cell electrophysiology and NO production [65] and the role of NO in weight loss [67]. In this study, we deduced the ameliorative effects of the permeability (selective or not) of the vascular wall from the increase in the phase angle and from the reduction of the controlled blood urea thirty minutes after finishing the dialysis session. To confirm our hypothesis, the impedance test demonstrates that PESF, after only one cycle, by positively modifying PA, acted on the functional and not on the biological patrimony of debilitated people. Hemodialysis sessions of elderly patients on hemodialysis are generally characterized by cardiovascular instability and short interruptions of purification sessions that do not allow to reach the pre-set weight for the end of dialysis (dry weight). The greater weight reduction in group A is connected, as well as with the greater vascular stability, with modified permeability of the endothelium.

Recently some authors have shown that the standardized phase angle, measured with bioelectrical impedance, is a prognostic biomarker independent of age, sex, degree of obesity (BMI) and comorbidity and is unfavorably correlated with mortality in inflamed patients (COVID-19). The patients most at risk are those with alterations of the vascular wall of the peripheral circulation (diabetes, hypertension, dyslipidemia or heart disease) and that the cell mass index and hydration can be identified with PA. Among the biomarkers related to metabolic health they cite albumin [65].

There is no doubt that the degree of "perceived health" affects both individual and social life and it is a necessary condition to achieve an increased family and social wellbeing.

The trained nurse, responsible for nursing, is in direct contact with the patient and can assess the level of satisfaction and the perceived quality of life. The questionnaire is a technical-scientific and management tool that can capture variations related to quality of life, when it is undermined through sickness, and the differences of perceived health. In our case, the nurse listened, informed, involved the patient and, as a close friend, assessed the need for treatment with him before and after the PDF. The self-administered questionnaires allow to measure the transversal differences in terms of quality of life among patients at a given time, within a group of patients or during a given observation period. Even when changes are small but related to individual disease states, groups of patients or functional areas, they can be noticed and reported.

The azotaemia is a waste product that derives from the metabolism of proteins. In patients with impaired renal efficiency, it accumulates first in *interstitium* and then in blood. The azotaemia also increases during an incongruous protein intake or during an increased protein catabolism. In patients with end-stage chronic renal failure, when the kidneys have become totally insufficient, it is necessary to intervene with replacement therapy to remove waste products. The artificial kidney reduces azotaemia during purification but, when ends dialysis, the catabolites accumulate again until the next dialysis session. Immediately after the end of the dialysis session, the passage of catabolites from the *interstitium* to the blood causes a new increase in blood urea. Only in group A we have observed statistically lower levels of azotaemia, after 30 minutes from the end of the dialysis session. We believe the result is unexpected, interesting and worthy of further research having observed it in uremic patients characterized

by marked protein catabolism. To date, there are no results regarding improvements of azotaemia levels obtained after a short period. We believe that PESF has influenced the metabolic process as evidenced by the increase in albumin and phase angle and is at the origin of less accumulation of nitrogenous products in the *interstitium*. Our interpretation is supported by the experimental results of other authors who attributed to PESF an increase in the supply of $O_2$ to peripheral tissues and an improvement in basal metabolism in obese individuals [19], greater vascular dynamism (vasomotion) in cardiac patients and accelerated healing of difficult ulcers in diabetic patients [20]. Circulating NO in elderly patients with chronic renal failure is reduced and is correlated with dysfunction of the endothelium and microcirculation. We believe that an increased in circulating NO in group A and although we are unable to specify whether the increase is endothelial (eNOS) or erythrocytic (rNOS), may be expression of an improved vascular performance. The increase in the percentage of albumin, the increase in PA and the positive relationship between albumin and PA are not justifiable if a PESF effect on the endothelial and microcirculation is not accepted. These results agree with those recorded on the best perceived quality of life.

The questionnaire highlighted improvements in physical functioning, on the limits attributed to the role of health (role limitation due to physical health), limitations attributed to the role of emotional problems (role limitations due to emotional problems) and health in general. (General Health).

These results agree with those observed and attributed to the improvement of the phase angle which is a sensitive indicator of health changes. Finally, we believe it is right to specify that we are aware of the effects of biofavonoids and phytotherapeutic substances including the Total Triterpenic Fraction (FTTCA) of Centella asiatica. Both categories of products act on the endothelium and reduce edema. We have previously dealt with arterial ulcers [20]. The derivatives of Centella Asiatica, applied locally, favour the healing of venous ulcers. Centella asiatica improves the supply of $O_2$ to tissues.

We are convinced that the results obtained are attributable to different effects that, although minimal but in a synergistic way, have all contributed to making tissue nutrition more efficient. Moreover, it is clear that "perceived pain" reduction in addition to the increase of physical activity releases from pain, it facilitates interpersonal relations and increases subjective wellbeing. The sense of less dependence reinforced the sense of adequacy of the subject, as documented. Although we cannot rule out that this phenomenon might have been induced by a better tissue perfusion, it is not easily interpreted by mechanisms that exclude tissue perfusion. In our future perspective, it will be necessary to investigate the impact of PESF at the endothelial level through, for example, the assessment of circulating endothelial cells (CEC), a reliable marker of disease activity in a variety of vascular disorders [68] and emerging markers of repair endothelial and activation/apoptosis [69]. The use of the technique in the long term requires further studies.

## Acknowledgments

We want to acknowledge dr. Mario Liani for his precious technical assistance and for critical reading of the manuscript.

## Author Contributions

**Conceptualization:** Carlo Velussi.

**Data curation:** Rossella Liani, Sara La Torre, Valentina Liani, Danilo D'Ettorre.

**Formal analysis:** Stefano Lattanzio.

**Investigation:** Rossella Liani, Rossano Di Luzio.

**Methodology:** Rossella Liani, Angela Melchiorre, Romina Tripaldi.

**Writing – original draft:** Rossella Liani.

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
