## [Decision Letter · Decision Letter 0]

20 Apr 2021

PONE-D-21-05955

Study of pulsed electrostatic field (PESF) in the perfusion of peripheral tissues: microangiopathy, nutrition and quality of perceiver life.

PLOS ONE

Dear Dr. Liani,

Thank you for submitting your manuscript to PLOS ONE. After careful consideration, we feel that it has merit but does not fully meet PLOS ONE’s publication criteria as it currently stands. Therefore, we invite you to submit a revised version of the manuscript that addresses the points raised during the review process.

We look forward to receiving your revised manuscript.

Kind regards,

Kanhaiya Singh, Ph.D

Academic Editor

PLOS ONE

Journal Requirements:

2. Please ensure you have included the registration number for the clinical trial referenced in the manuscript.

3. Thank you for stating the following financial disclosure: 'NO'

4. Thank you for stating the following in your Competing Interests section: 'NO'

a. Please complete your Competing Interests statement to state any Competing Interests. If you have no competing interests, please state "The authors have declared that no competing interests exist.", as detailed online in our guide for authors at http://journals.plos.org/plosone/s/submit-now

Additional Editor Comments:

Although reviewers found this study interesting, they are concerned about the statistical interpretation of the results. They also found that the authors have made certain assumptions without proper data support. The manuscript at present needs through revision to address the comments and criticisms about technical aspects of the analyses.

Reviewers' comments:

Reviewer's Responses to Questions

**Comments to the Author**

1. Is the manuscript technically sound, and do the data support the conclusions?

Reviewer #1: No

Reviewer #2: Yes

Reviewer #3: No

2. Has the statistical analysis been performed appropriately and rigorously? 

Reviewer #1: Yes

Reviewer #2: No

Reviewer #3: No

3. Have the authors made all data underlying the findings in their manuscript fully available?

Reviewer #1: No

Reviewer #2: Yes

Reviewer #3: Yes

4. Is the manuscript presented in an intelligible fashion and written in standard English?

Reviewer #1: No

Reviewer #2: Yes

Reviewer #3: Yes

5. Review Comments to the Author

Reviewer #1: The authors present a study on the effect of PESF on elderly patients undergoing hemodialysis. This was a long and involved study that has some interesting observations but for reasons listed below would require significant modification to be accepted for publication.

1. The entire manuscript needs to be checked for grammar and composition. There are significant errors in sentence structure.

2. Are the authors following a standard guideline for clinical research studies - like STROBE - since this appears to be observational? This is unclear.

3. Authors make a lot of assumptions based on very little data that is not clearly laid out. Details below:

a. Figure 1 - very hard to follow - the authors use the words no change (gp B) and homogeneity (gp A) - what is the difference? The figure shows 5 components in Gp B panel and 6 components in Gp A panel. Also note - the word used by authors is componenti - which needs to be corrected. The authors cannot expect a viewer of that figure to know at-a-glance what the difference is pre and post treatment in groups A and B. There must be a clearer representation of the exact differences pointed out or shown in a different format.

b. Authors describe measuring azotaemia levels - but where is the figure to show actual numbers? Only statistics are descriptively included.

b. Although the authors are comparing two cohorts that are age-matched and matched for gender - there is insufficient demographic information to be clear about how other underlying conditions could impact the outcomes of their studies - not discussed at all.

c. It is unclear how soon after the treatment the questionnaire was applied - so is the effect in group A short term or long term?

d. Insufficient details in methodology: What were the simulated and actual test parameters? The authors reference a paper but there needs to be a brief summary of what was applied in this study.

e. The authors point out that NO levels are increased in Group A after treatment. Yet they do not discuss what the significance of this might be? Are the levels physiologically relevant to make an impact?

f. In all other data tables - the pre treatment levels were comparable between group A and B. However in table 3 - group B pretreatment seems quite high (equation (ln) column) compared to the same cohort in group A - can the authors explain that difference?

g. The discussion section is very wordy and reads like a general review - instead of specific and focused comments based on the actual observations they made.

Reviewer #2: Liani et al., studied the effect of of pulsed electrostatic field (PESF) in the perfusion of peripheral tissues: microangiopathy, nutrition and quality of perceiver life. The manuscript has raised some major concerns.

1. In Table 2 the difference in values in pre-and post of group A is not substantial (though the p value is less than 0.05). The SD is large and the number of samples is not that big. Same type of differences are seen in Group B but not significant. The authors must show individual data pints in the form of graph and confirm the p values.

2. The title claims nutrition which is not at all mentioned in the results or discussion

3. Figure 1 is not clear.

Reviewer #3: The study is interesting, was rigorously done for 5 years, the perfusion of peripheral tissues employing employing pulsed electrostatic field was novel, however they were significant gaps, that authors could address these following gaps

1. There are literature evidences suggesting that there is increase in nitric oxide levels & sensitivity with weight loss, authors in the table 2 talk about significant difference in weight loss (p = 0.007), the control group had no weight loss, there is no discussion how the weight loss impacts (given in the table 2), the nitric oxide levels in case of PESF treated and untreated groups, Authors could include few lines in discussion if there is any positive co-relation between weight loss, PESF, and Nitric oxide levels with appropriate references.

2. There is no clear elaboration on why there are no molecular markers included in the study for determining the microangiopathy recovery or progression. Authors could possible include one blood marker to indicate is due recovery due to synergistic effect of PESF other than NO levels.

3. The study would have been more interesting, if the authors could include any other treatment of microangiopathy (example but not limited) total triterpenic fraction of Centella asiatica (TTFCA) as another control group to determine the how positive is PESF bringing the life changes in microangiopathy

6. PLOS authors have the option to publish the peer review history of their article (what does this mean?). If published, this will include your full peer review and any attached files.

Reviewer #1: No

Reviewer #2: No

Reviewer #3: No

---

## [Author Response · Author response to Decision Letter 0]

25 Jun 2021

Dear Reviewers, 

thanks for constructive comments which made it possible to improve the paper.

1. The entire manuscript needs to be checked for grammar and composition. There are significant errors in sentence structure.

We have revised grammar composition

2. Are the authors following a standard guideline for clinical research studies - like STROBE - since this appears to be observational? This is unclear.

Our study is an observational and retrospective clinical report using the electronic archive present in our Operating Unit in according of STROBE criteria

3. Authors make a lot of assumptions based on very little data that is not clearly laid out. Details below:

a. Figure 1 - very hard to follow - the authors use the words no change (gp B) and homogeneity (gp A) - what is the difference? The figure shows 5 components in Gp B panel and 6 components in Gp A panel. Also note - the word used by authors is componenti - which needs to be corrected. The authors cannot expect a viewer of that figure to know at-a-glance what the difference is pre and post treatment in groups A and B. There must be a clearer representation of the exact differences pointed out or shown in a different format.

We have clarified Figure 1 and correct word mistake.

b. Authors describe measuring azotaemia levels - but where is the figure to show actual numbers? Only statistics are descriptively included.

We have insert azotaemia numbers

b. Although the authors are comparing two cohorts that are age-matched and matched for gender - there is insufficient demographic information to be clear about how other underlying conditions could impact the outcomes of their studies - not discussed at all.

The two groups were uniform and showed no noteworthy statistical differences

c. It is unclear how soon after the treatment the questionnaire was applied - so is the effect in group A short term or long term?

The effects were evaluated just after the treatment, and it would be interesting to re-evaluate them in the long term

d. Insufficient details in methodology: What were the simulated and actual test parameters? The authors reference a paper but there needs to be a brief summary of what was applied in this study.

We have implemented the methods even though there are references from our previous studies where the treatment details are present

e. The authors point out that NO levels are increased in Group A after treatment. Yet they do not discuss what the significance of this might be? Are the levels physiologically relevant to make an impact?

f. In all other data tables - the pre treatment levels were comparable between group A and B. However in table 3 - group B pretreatment seems quite high (equation (ln) column) compared to the same cohort in group A - can the authors explain that difference?

The pre of group B is not significantly different from group A

g. The discussion section is very wordy and reads like a general review - instead of specific and focused comments based on the actual observations they made.

Thanks we have remodeled the discussion

Reviewer #2: Liani et al., studied the effect of of pulsed electrostatic field (PESF) in the perfusion of peripheral tissues: microangiopathy, nutrition and quality of perceiver life. The manuscript has raised some major concerns.

1. In Table 2 the difference in values in pre-and post of group A is not substantial (though the p value is less than 0.05). The SD is large and the number of samples is not that big. Same type of differences are seen in Group B but not significant. The authors must show individual data pints in the form of graph and confirm the p values.

We confirm the statistics reported

2. The title claims nutrition which is not at all mentioned in the results or discussion

We have insert nutrition profile in discussion

3. Figure 1 is not clear.

We have clarified Figure 1 and correct word mistake.

Reviewer #3: The study is interesting, was rigorously done for 5 years, the perfusion of peripheral tissues employing employing pulsed electrostatic field was novel, however they were significant gaps, that authors could address these following gaps

1. There are literature evidences suggesting that there is increase in nitric oxide levels & sensitivity with weight loss, authors in the table 2 talk about significant difference in weight loss (p = 0.007), the control group had no weight loss, there is no discussion how the weight loss impacts (given in the table 2), the nitric oxide levels in case of PESF treated and untreated groups, Authors could include few lines in discussion if there is any positive co-relation between weight loss, PESF, and Nitric oxide levels with appropriate references.

We have introduced this aspect into the discussion

2. There is no clear elaboration on why there are no molecular markers included in the study for determining the microangiopathy recovery or progression. Authors could possible include one blood marker to indicate is due recovery due to synergistic effect of PESF other than NO levels.

Thanks for the tip we will keep it in consideration for future studies 

3. The study would have been more interesting, if the authors could include any other treatment of microangiopathy (example but not limited) total triterpenic fraction of Centella asiatica (TTFCA) as another control group to determine the how positive is PESF bringing the life changes in microangiopathy

We excluded patients with major arteries occlusion, valued with doppler test and patients who used: polyphenols, antioxidants for their effects on the cellular signal; phytotherapics (eg. Centella asiatica extracts) for their known effects on endothelial cells; anabolic drugs for their effects on metabolism; and psychotropic drugs. we believe it is right to specify that we are aware of the effects of biofavonoids and phytotherapeutic substances including the Total Triterpenic Fraction (FTTCA) of Centella asiatica. Both categories of products act on the endothelium and reduce edema. We have previously dealt with arterial ulcers[20]. The derivatives of Centella Asiatica, applied locally, favour the healing of venous ulcers. Centella asiatica improves the supply of O2 to tissues.

---

## [Decision Letter · Decision Letter 1]

6 Dec 2021

PONE-D-21-05955R1

Study of pulsed electrostatic field (PESF) in the perfusion of peripheral tissues: microangiopathy, nutrition and quality of perceiver life.

PLOS ONE

Dear Dr. Liani,

Thank you for submitting your manuscript to PLOS ONE. After careful consideration, we feel that it has merit but does not fully meet PLOS ONE’s publication criteria as it currently stands. Therefore, we invite you to submit a revised version of the manuscript that addresses the points raised during the review process.

We look forward to receiving your revised manuscript.

Kind regards,

Kanhaiya Singh, Ph.D

Academic Editor

PLOS ONE

Journal Requirements:

Additional Editor Comments:

Although reviewers have acknowledge the significant efforts of authors to revise this manuscript, they have recommended some more changes in order to make this study robust for publication. Please include individual data points for conclusions drawn for Table 2. Reviewer 3 comments are mentioned below:

Reviewer 3:

1. There are literature evidences suggesting that there is increase in nitric oxide levels

& sensitivity with weight loss, authors in the table 2 talk about significant difference in

weight loss (p = 0.007), the control group had no weight loss, there is no discussion

how the weight loss impacts (given in the table 2), the nitric oxide levels in case of

PESF treated and untreated groups, Authors could include few lines in discussion if

there is any positive co-relation between weight loss, PESF, and Nitric oxide levels with

appropriate references.

Response: We have introduced this aspect into the discussion

Reviewer 3: Could you please elaborate the discussion further about positive co-relation between weight loss, PESF, and Nitric oxide levels with appropriate references.

2. There is no clear elaboration on why there are no molecular markers included in the

study for determining the microangiopathy recovery or progression. Authors could

possible include one blood marker to indicate is due recovery due to synergistic effect

of PESF other than NO levels.

Response: Thanks for the tip we will keep it in consideration for future studies

Reviewer 3: Could you please add a section on how circulating endothelial cells (CECs) are a reliable marker of disease activity in a variety of vascular disorders. If possible an experiment to determine the increase in count of endothelial cell post PESF.

3. The study would have been more interesting, if the authors could include any other

treatment of microangiopathy (example but not limited) total triterpenic fraction of

Centella asiatica (TTFCA) as another control group to determine the how positive is

PESF bringing the life changes in microangiopathy

Response: We excluded patients with major arteries occlusion, valued with doppler test and

patients who used: polyphenols, antioxidants for their effects on the cellular signal;

phytotherapics (eg. Centella asiatica extracts) for their known effects on endothelial

cells; anabolic drugs for their effects on metabolism; and psychotropic drugs. we

believe it is right to specify that we are aware of the effects of biofavonoids and

phytotherapeutic substances including the Total Triterpenic Fraction (FTTCA) of

Centella asiatica. Both categories of products act on the endothelium and reduce

edema. We have previously dealt with arterial ulcers[20]. The derivatives of Centella

Asiatica, applied locally, favour the healing of venous ulcers. Centella asiatica

improves the supply of O2 to tissues.

Reviewer 3: Could you please include a patient group subjected through a different mode of treatment for microangiopathy as a positive control group

Reviewers' comments:

Reviewer's Responses to Questions

**Comments to the Author**

1. If the authors have adequately addressed your comments raised in a previous round of review and you feel that this manuscript is now acceptable for publication, you may indicate that here to bypass the “Comments to the Author” section, enter your conflict of interest statement in the “Confidential to Editor” section, and submit your "Accept" recommendation.

Reviewer #1: (No Response)

Reviewer #2: (No Response)

Reviewer #3: (No Response)

2. Is the manuscript technically sound, and do the data support the conclusions?

Reviewer #1: Partly

Reviewer #2: Yes

Reviewer #3: (No Response)

3. Has the statistical analysis been performed appropriately and rigorously? 

Reviewer #1: Yes

Reviewer #2: No

Reviewer #3: (No Response)

4. Have the authors made all data underlying the findings in their manuscript fully available?

Reviewer #1: (No Response)

Reviewer #2: Yes

Reviewer #3: (No Response)

5. Is the manuscript presented in an intelligible fashion and written in standard English?

Reviewer #1: (No Response)

Reviewer #2: Yes

Reviewer #3: (No Response)

6. Review Comments to the Author

Reviewer #1: The reviewers have made a good effort to improve their manuscript. Few points that need addressing in order to accept

1. While the authors did clean up most of their grammatical issues, there are still some that need revising. If you are able to get a professional english editing service to help, that might be very useful. Examples:

a. Patients treated with PESF were improved circulating nitric oxide levels - should read patients treated with PESF 'showed' improved...

b. improper and incomplete internal editing: In elders with T2DM and chronic kidney disease (CKD), without

macrovascular precedents, microangiopathy effects the early atrophy of the brain matter of the frontal

and temporal lobes and it affects memory, the executive functions and ultimately the behaviourFare

clic o toccare qui per immettere il testo.[12].

c. present in our Operating Unit in according of STROBE criteria - should read - present in our operating unit in "accordance with or following" STROBE criteria.

2. No response given to: e. The authors point out that NO levels are increased in Group A after treatment. Yet

they do not discuss what the significance of this might be? Are the levels

physiologically relevant to make an impact?

Perhaps the authors did make edits to account for this but it is not immediately clear.

3. I don't see figure legends included for the 2 figures.

4. Table cannot be shortened to Tab. Please write out the word Table.

I think with these edits, the manuscript may be ok to be accepted for publication.

Reviewer #2: In Table 2 the difference in values in pre-and post of group A is not substantial

(though the p value is less than 0.05). The SD is large and the number of samples is

not that big. Same type of differences are seen in Group B but not significant. The

authors MUST SHOW individual data pints in the form of graph and confirm the p values.

Reviewer #3: (No Response)

7. PLOS authors have the option to publish the peer review history of their article (what does this mean?). If published, this will include your full peer review and any attached files.

Reviewer #1: No

Reviewer #2: No

Reviewer #3: No

---

## [Author Response · Author response to Decision Letter 1]

8 Feb 2022

Date: Dec 06 2021 07:59AM

To: "Rossella Liani" rossellaliani@yahoo.it

From: "PLOS ONE" plosone@plos.org

Subject: PLOS ONE Decision: Revision required [PONE-D-21-05955R1]

PONE-D-21-05955R1

Study of pulsed electrostatic field (PESF) in the perfusion of peripheral tissues: microangiopathy, nutrition and quality of perceiver life.

PLOS ONE

Additional Editor Comments:

Although reviewers have acknowledge the significant efforts of authors to revise this manuscript, they have recommended some more changes in order to make this study robust for publication. Please include individual data points for conclusions drawn for Table 2. Reviewer 3 comments are mentioned below:

Reviewer 3:

1. There are literature evidences suggesting that there is increase in nitric oxide levels

& sensitivity with weight loss, authors in the table 2 talk about significant difference in

weight loss (p = 0.007), the control group had no weight loss, there is no discussion

how the weight loss impacts (given in the table 2), the nitric oxide levels in case of

PESF treated and untreated groups, Authors could include few lines in discussion if

there is any positive co-relation between weight loss, PESF, and Nitric oxide levels with

appropriate references.

Response: We have introduced this aspect into the discussion

Reviewer 3: Could you please elaborate the discussion further about positive co-relation between weight loss, PESF, and Nitric oxide levels with appropriate references.

Response: We have introduced this aspect into the discussion

2. There is no clear elaboration on why there are no molecular markers included in the

study for determining the microangiopathy recovery or progression. Authors could

possible include one blood marker to indicate is due recovery due to synergistic effect

of PESF other than NO levels.

Response: Thanks for the tip we will keep it in consideration for future studies

Reviewer 3: Could you please add a section on how circulating endothelial cells (CECs) are a reliable marker of disease activity in a variety of vascular disorders. If possible an experiment to determine the increase in count of endothelial cell post PESF.

Response: We have introduced this aspect into the discussion but the study is over and the CEC are valued on fresh sample (DOI: 10.1016/j.jim.2012.03.007; DOI: 10.1038/s41598-021-88941-x)

3. The study would have been more interesting, if the authors could include any other

treatment of microangiopathy (example but not limited) total triterpenic fraction of

Centella asiatica (TTFCA) as another control group to determine the how positive is

PESF bringing the life changes in microangiopathy

Response: We excluded patients with major arteries occlusion, valued with doppler test and

patients who used: polyphenols, antioxidants for their effects on the cellular signal;

phytotherapics (eg. Centella asiatica extracts) for their known effects on endothelial

cells; anabolic drugs for their effects on metabolism; and psychotropic drugs. we

believe it is right to specify that we are aware of the effects of biofavonoids and

phytotherapeutic substances including the Total Triterpenic Fraction (FTTCA) of

Centella asiatica. Both categories of products act on the endothelium and reduce

edema. We have previously dealt with arterial ulcers[20]. The derivatives of Centella

Asiatica, applied locally, favour the healing of venous ulcers. Centella asiatica

improves the supply of O2 to tissues.

Reviewer 3: Could you please include a patient group subjected through a different mode of treatment for microangiopathy as a positive control group

Response: Thanks for the tip we will keep it in consideration for future studies, but the study is over

Reviewer's Responses to Questions

Comments to the Author

1. If the authors have adequately addressed your comments raised in a previous round of review and you feel that this manuscript is now acceptable for publication, you may indicate that here to bypass the “Comments to the Author” section, enter your conflict of interest statement in the “Confidential to Editor” section, and submit your "Accept" recommendation.

Reviewer #1: (No Response)

Reviewer #2: (No Response)

Reviewer #3: (No Response)

2. Is the manuscript technically sound, and do the data support the conclusions?

Reviewer #1: Partly

Reviewer #2: Yes

Reviewer #3: (No Response)

3. Has the statistical analysis been performed appropriately and rigorously? 

Reviewer #1: Yes

Reviewer #2: No

Reviewer #3: (No Response)

4. Have the authors made all data underlying the findings in their manuscript fully available?

Reviewer #1: (No Response)

Reviewer #2: Yes

Reviewer #3: (No Response)

5. Is the manuscript presented in an intelligible fashion and written in standard English?

Reviewer #1: (No Response)

Reviewer #2: Yes

Reviewer #3: (No Response)

6. Review Comments to the Author

Reviewer #1: The reviewers have made a good effort to improve their manuscript. Few points that need addressing in order to accept

1. While the authors did clean up most of their grammatical issues, there are still some that need revising. If you are able to get a professional english editing service to help, that might be very useful. Examples:

a. Patients treated with PESF were improved circulating nitric oxide levels - should read patients treated with PESF 'showed' improved...

b. improper and incomplete internal editing: In elders with T2DM and chronic kidney disease (CKD), without

macrovascular precedents, microangiopathy effects the early atrophy of the brain matter of the frontal

and temporal lobes and it affects memory, the executive functions and ultimately the behaviourFare

clic o toccare qui per immettere il testo.[12].

c. present in our Operating Unit in according of STROBE criteria - should read - present in our operating unit in "accordance with or following" STROBE criteria.

Thanks for the suggestions, we have corrected in the text

2. No response given to: e. The authors point out that NO levels are increased in Group A after treatment. Yet

they do not discuss what the significance of this might be? Are the levels

physiologically relevant to make an impact?

Perhaps the authors did make edits to account for this but it is not immediately clear.

Response: We have improved this aspect into the discussion

3. I don't see figure legends included for the 2 figures.

Response: Thanks. We have included figure legends for the 2 figures

 4. Table cannot be shortened to Tab. Please write out the word Table.

Response: Thanks. 

I think with these edits, the manuscript may be ok to be accepted for publication.

Reviewer #2: In Table 2 the difference in values in pre-and post of group A is not substantial

(though the p value is less than 0.05). The SD is large and the number of samples is

not that big. Same type of differences are seen in Group B but not significant. The

authors MUST SHOW individual data pints in the form of graph and confirm the p values.

Response: Thanks, but the p value is at most equal to 0.021 in Table 2, and not near to 0.05, full significance P value 

Reviewer #3: (No Response)

7. PLOS authors have the option to publish the peer review history of their article (what does this mean?). If published, this will include your full peer review and any attached files.

Do you want your identity to be public for this peer review? For information about this choice, including consent withdrawal, please see our Privacy Policy.

Reviewer #1: No

Reviewer #2: No

Reviewer #3: No

---

## [Decision Letter · Decision Letter 2]

28 Feb 2022

PONE-D-21-05955R2Study of pulsed electrostatic field (PESF) in the perfusion of peripheral tissues: microangiopathy, nutrition and quality of perceiver lifePLOS ONE

Dear Dr. Liani,

Thank you for submitting your manuscript to PLOS ONE. After careful consideration, we feel that it has merit but does not fully meet PLOS ONE’s publication criteria as it currently stands. Therefore, we invite you to submit a revised version of the manuscript that addresses the points raised during the review process.

We look forward to receiving your revised manuscript.

Kind regards,

Kanhaiya Singh, Ph.D

Academic Editor

PLOS ONE

Journal Requirements:

Additional Editor Comments (if provided):

Please respond to the query raised by Reviewer 2. They wish to see the individual data points.

Reviewers' comments:

Reviewer's Responses to Questions

**Comments to the Author**

1. If the authors have adequately addressed your comments raised in a previous round of review and you feel that this manuscript is now acceptable for publication, you may indicate that here to bypass the “Comments to the Author” section, enter your conflict of interest statement in the “Confidential to Editor” section, and submit your "Accept" recommendation.

Reviewer #1: All comments have been addressed

Reviewer #2: (No Response)

Reviewer #3: All comments have been addressed

2. Is the manuscript technically sound, and do the data support the conclusions?

Reviewer #1: (No Response)

Reviewer #2: Yes

Reviewer #3: Yes

3. Has the statistical analysis been performed appropriately and rigorously? 

Reviewer #1: Yes

Reviewer #2: Yes

Reviewer #3: Yes

4. Have the authors made all data underlying the findings in their manuscript fully available?

Reviewer #1: Yes

Reviewer #2: (No Response)

Reviewer #3: Yes

5. Is the manuscript presented in an intelligible fashion and written in standard English?

Reviewer #1: Yes

Reviewer #2: Yes

Reviewer #3: Yes

6. Review Comments to the Author

Reviewer #1: (No Response)

Reviewer #2: I have repeatedly been asking for the individual data points in Table 2. The authors MUST SHOW individual data pints in the form of graph or provide the excel sheet for us to review. THIS HASN'T BEEN ADDRESSED.

Reviewer #3: (No Response)

7. PLOS authors have the option to publish the peer review history of their article (what does this mean?). If published, this will include your full peer review and any attached files.

Reviewer #1: No

Reviewer #2: No

Reviewer #3: No

---

## [Author Response · Author response to Decision Letter 2]

30 Mar 2022

Reviewer #2: I have repeatedly been asking for the individual data points in Table 2. The authors MUST SHOW individual data pints in the form of graph or provide the excel sheet for us to review. THIS HASN'T BEEN ADDRESSED.

Dear Reviewer #2: we have replaced table 2 with the new figure 2 (dot plot) where the single points are represented and repeated the statistic using the most appropriate Mann Whitney non-parametric test.

---

## [Decision Letter · Decision Letter 3]

3 May 2022

Study of pulsed electrostatic field (PESF) in the perfusion of peripheral tissues: microangiopathy, nutrition and quality of perceiver life

PONE-D-21-05955R3

Dear Dr. Liani,

We’re pleased to inform you that your manuscript has been judged scientifically suitable for publication and will be formally accepted for publication once it meets all outstanding technical requirements.

Kind regards,

Kanhaiya Singh, Ph.D

Academic Editor

PLOS ONE

Additional Editor Comments (optional):

Reviewers' comments:

Reviewer's Responses to Questions

**Comments to the Author**

1. If the authors have adequately addressed your comments raised in a previous round of review and you feel that this manuscript is now acceptable for publication, you may indicate that here to bypass the “Comments to the Author” section, enter your conflict of interest statement in the “Confidential to Editor” section, and submit your "Accept" recommendation.

Reviewer #2: All comments have been addressed

2. Is the manuscript technically sound, and do the data support the conclusions?

Reviewer #2: Yes

3. Has the statistical analysis been performed appropriately and rigorously? 

Reviewer #2: Yes

4. Have the authors made all data underlying the findings in their manuscript fully available?

Reviewer #2: Yes

5. Is the manuscript presented in an intelligible fashion and written in standard English?

Reviewer #2: Yes

6. Review Comments to the Author

Reviewer #2: All comments addressed satisfactorily. This manuscript is now ready to be accepted in its present form.

7. PLOS authors have the option to publish the peer review history of their article (what does this mean?). If published, this will include your full peer review and any attached files.

Reviewer #2: No

---

## [Editor Report · Acceptance letter]

10 May 2022

PONE-D-21-05955R3 

Study of pulsed electrostatic field (PESF) in the perfusion of peripheral tissues: microangiopathy, nutrition and quality of perceiver life. 

Dear Dr. Liani:

I'm pleased to inform you that your manuscript has been deemed suitable for publication in PLOS ONE. Congratulations! Your manuscript is now with our production department. 

Kind regards, 

on behalf of

Dr. Kanhaiya Singh 

Academic Editor

PLOS ONE